# Consistent Instance Classification for Unsupervised Representation Learning

## Abstract

In this paper, we address the problem of learning the representations from images without human annotations. We study the instance classification solution, which regards each instance as a category, and improve the optimization and feature quality. The proposed consistent instance classification (ConIC) approach simultaneously optimizes the classification loss and an additional consistency loss explicitly penalizing the feature dissimilarity between the augmented views from the same instance. The benefit of optimizing the consistency loss is that the learned features for augmented views from the same instance are more compact and accordingly the classification loss optimization becomes easier, thus boosting the quality of the learned representations. This differs from InstDisc (Wu et al., 2018) and MoCo (He et al., 2019; Chen et al., 2020c) that use an estimated prototype as the classifier weight to ease the optimization. Different from SimCLR (Chen et al., 2020b) that directly compares different instances, our approach does not require large batch size. Experimental results demonstrate competitive performance for linear evaluation and better performance than InstDisc, MoCo and SimCLR at downstream tasks, such as detection and segmentation, as well as competitive or superior performance compared to other methods with stronger training setting.

## 1 Introduction

Learning good representations from unlabeled images is a land-standing and challenging problem. The mainstream methods include: generative modeling (Hinton et al., 2006; Kingma & Welling, 2014), colorization (Zhang et al., 2016), transformation or spatial relation prediction (Doersch et al., 2015; Noroozi & Favaro, 2016; Gidaris et al., 2018), and discriminative methods, such as instance classification (Dosovitskiy et al., 2016; He et al., 2019), and contrastive learning (Chen et al., 2020b).

The instance discrimination methods show promising performance for downstream tasks. There are two basic objectives that are optimized (He et al., 2019; Chen et al., 2020b; Yu et al., 2020; Wang & Isola, 2020): contraction and separation. Contraction means that the features of the augmented views from the same instance should be as close as possible. Separation means that the features of the augmented views from one instance should lie in a region different from other instances.

The instance classification framework, such as InstDisc (Wu et al., 2018), and MoCo (He et al., 2019; Chen et al., 2020c), adopts a prototype-based classifier, where the prototype is estimated as the moving average of the corresponding features of previous epoches Wu et al. (2018) or as the output of an moving-average network He et al. (2019); Chen et al. (2020c). The prototype-based schemes ease the optimization of the classification loss in the challenging case that there is over one million categories. BYOL (Grill et al., 2020) computes the prototype in a way similar to MoCo, and only aligns the feature of augmented views with its prototype leaving the separation objective implicitly optimized. The prototype, computed from a single view rather than many views and from networks with different parameters, might not be reliable enough, making the contraction and separation optimization quality not guaranteed.

The contrastive learning framework[1], such as SimCLR (Chen et al., 2020b) and Ye et al. (2019), simultaneously maximizes the similarities between each view pair from the same instance and min-

---

[1]InstDisc (Wu et al., 2018) and MoCo (He et al., 2019; Chen et al., 2020c) are also closely related to contrastive learning and are regarded as contrastive learning methods by some researchers.

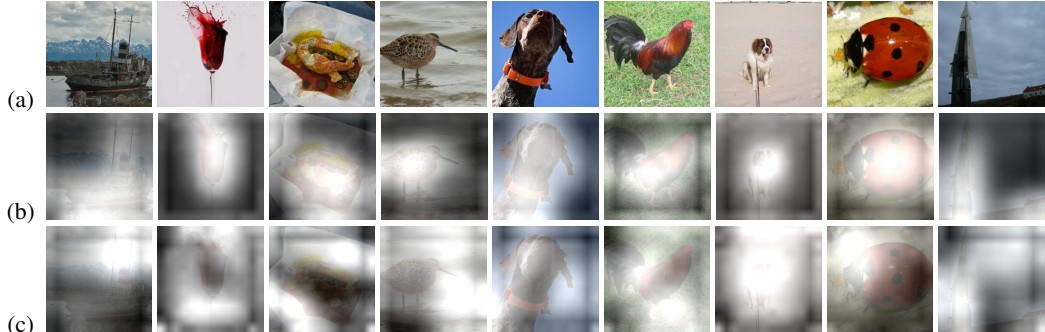

Figure 1: Visualizing the activation maps. (a) input image, (b) activation maps from our approach, (c) activation maps from only optimizing the classification loss. One can see that our approach (b) tends to more focus on the textured region.

imizes the similarities between the view pair from different instances. This framework directly compares the feature of one view to a different view other than to a prototype, avoiding the unreliability of the prototype estimation. It, however, requires large batch size for each SGD iteration to compare enough number of negative instances for imposing the separation constraint[2], increasing the difficulty in large batch training.

We propose a simple unsupervised representation learning approach, consistent instance classification (ConIC), to improve the optimization and feature quality. Our approach jointly minimizes two losses: instance classification loss and consistency loss. The instance classification loss is formulated by regarding each instance as a category. Its optimization encourages that different instances lie in different regions. The consistency loss is formulated to directly compares the features of the augmented views from the same instance and encourages high similarity between them.

One benefit from the consistency optimization is to directly and explicitly make the features of the same instances compact and thus to accelerate the optimization of the classification loss. This is different from Wu et al. (2018), He et al. (2019), heuristically estimating the classifier weights using the prototypes and does not suffer from the prototype estimation reliability issue. On the other hand, our approach does not rely on large batch training, that is essential for SimCLR (Chen et al., 2020b), because the whole loss in our formulation can be decomposed as a sum of components each of which only depends on one instance.

Furthermore, we observed that jointly optimizing the consistency and classification losses leads to that the representation is more focused on the textured region, as shown in Figure 1. This implies that the learned representation is more capable of characterizing the objects, and thus potentially more helpful for downstream tasks like object detection and segmentation.

We demonstrate the effectiveness of our approach in unsupervised representation learning on ImageNet. Our approach achieves competitive performance under the linear evaluation protocol. When finetuned on downstream tasks, such as object detection on VOC, object detection and instance segmentation on COCO, instance segmentation on Cityscapes and LVIS, as well as semantic segmentation on Citeyscapes, COCO Stuff, ADE and VOC, our approach performs better than InstDisc, MoCo and SimCLR, and competitively or superior compared to other methods with stronger training setting (e.g., InfoMin and SwAV).

## 2 RELATED WORK

**Generative approaches.** Generative models, such as auto-encoders (Hinton et al., 2006; Kingma & Welling, 2014; Vincent et al., 2008), context encoders (Pathak et al., 2016), GANs (Donahue & Simonyan, 2019), and GPTs (Chen et al., 2020a), learn an unsupervised representation by faithfully reconstructing the pixels. Later self-supervised models, such as colorization (Zhang et al., 2016) and split-brain encoders (Zhang et al., 2017), improve generative models by withholding some part of the data and predicting it.

---

[2]We will show one possible reason that it requires large batch.

**Spatial relation prediction.** The representation is learned by solving pretext tasks related to image patch spatial relation prediction, such as predicting the spatial relation between two patches sampled from an image, e.g., a patch is on the left of another patch, (Doersch et al., 2015); solving Jigsaw Puzzles and determining the spatial configuration for the shuffled (typically 9) patches (Noroozi & Favaro, 2016); and predicting the rotation (Gidaris et al., 2018).

**Instance classification.** Exemplar-CNN (Dosovitskiy et al., 2016) regards the views formed by augmenting each instance as a class, and formulates an instance classification problem. InstDisc (Wu et al., 2018), MoCo (He et al., 2019), CMC (Tian et al., 2019) and PIRL (Misra & van der Maaten, 2019) generalize exemplar-CNN by heuristically estimating the classifier weights using prototypes for easing the optimization. Our proposed approach follows the instance classification approach, and exploit an additional consistency loss to help optimization.

**Instance clustering.** Rather than regarding each instance as a category, the instance clustering frameworks (Caron et al., 2018; 2019; 2020; Asano et al., 2020; Huang et al., 2019; Xie et al., 2016; Yan et al., 2020; Yang et al., 2016) learn representations in which the instances are well optimized. DeepCluster (Caron et al., 2018) simply adopt the $k$-means clustering method by simultaneously optimizing the network parameters, and uses $k$-means assignments as pseudo-labels to learn representations. SwAV (Caron et al., 2020) simultaneously clusters the data while enforcing consistency between cluster assignments for different views of the same instance.

**Contrastive learning.** Contrastive predictive coding (van den Oord et al., 2018; Hénaff et al., 2019) predicts the representations of patches below a certain position from those above it by optimizing contrastive loss. DIM (Hjelm et al., 2019) and ANDIM (Bachman et al., 2019) achieves global-to-local/local-to-neighbor patch representation prediction (overlapping) across augmented views using the contrastive loss.

The contrastive learning framework (Ye et al., 2019; Chen et al., 2020b) formulates a contrastive loss encouraging the high similarity between the augmented views from the same instance, and low similarity between the instance and other instances. Wang & Isola (2020) presents a novel formulation based on two measures: alignment and uniformity, and shows that it is an alternative of contrastive loss. Yu et al. (2020) connects contractive and contrastive learning, cross-entropy, and so on, and provides theoretical guarantees for learning diverse and discriminative features.

**Consistency in semi-supervised learning.** Consistency regularization, enforcing the similarity between the predictions or features of different views for the same unlabeled instance, has been widely applied in semi-supervised learning, such as Π Model (Laine & Aila, 2017), Temporal Ensemble (Laine & Aila, 2017), and Mean Teacher (Tarvainen & Valpola, 2017). We exploit the consistency loss to help optimize the classification loss for unsupervised representation learning.

## 3 APPROACH

Given a set of image instances without any labels, $\mathcal{I} = \{\mathbf{I}_1, \mathbf{I}_2, \ldots, \mathbf{I}_N\}$, the goal is to learn a feature extractor (a neural network) $\mathbf{x} = f(\mathbf{I})$. The discrimination approach expands each image $\mathbf{I}_n$ to a set of augmented views $\{\mathbf{I}_n^1, \mathbf{I}_n^2, \ldots, \mathbf{I}_n^K\}$, and formulates the problem in a way that the features of the augmented views of each instance are similar (contraction) and the features of different instances are distributed separately (separation). In the following, we first review three related instance classification methods, then we introduce our approach and present the analysis.

### 3.1 INSTANCE CLASSIFICATION

**Exemplar CNN.** Exemplar-CNN (Dosovitskiy et al., 2016) formulates unsupervised representation learning as an instance classification problem. The augmented views from one instance are regarded as one category, and the augmented views from different instances are regarded as different categories. The softmax loss is used and written for the $k$th view of the $n$th instance:

$$\ell_s(\mathbf{x}_n^k) = -\log \frac{e^{\mathbf{w}_n^\top \tilde{\mathbf{x}}_n^k / \tau}}{\sum_{j=1}^N e^{\mathbf{w}_j^\top \tilde{\mathbf{x}}_n^k / \tau}}, \tag{1}$$

where $\tau$ is the temperature. Exemplar-CNN uses the standard backpropagation algorithm to learn the network $f(\cdot)$ and the classification weights $\{\mathbf{w}_1, \mathbf{w}_2, \ldots, \mathbf{w}_N\}$.

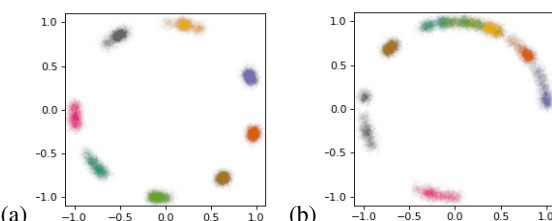 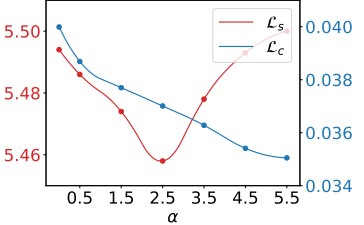

Figure 2: Visualizing learned feature distributions for 2D toy examples. Each color corresponds to augmented views of the same instance. (a) jointly optimize the consistency and classification losses. (b) only optimize the classification loss.

Figure 3: Illustration of the final classification loss and consistency loss values in training.

**InstDisc.** The InstDisc approach (Wu et al., 2018) optimizes the network parameters, and heuristically estimates the classifier weights $\{\mathbf{w}_1, \mathbf{w}_2, \ldots, \mathbf{w}_N\}$ in each epoch using a feature moving average scheme, i.e., compute the exponential averages of the features of the corresponding instances (stored in a memory bank) in the previous epochs. The heuristic weight estimation scheme eases the network optimization.

**MoCo.** MoCo (He et al., 2019) instead adopts a network moving average scheme. In each SGD iteration, MoCo updates a momentum network whose parameters are moving average of the previous network parameters. It computes the features from the momentum network as the classifier weights, which are further maintained by a queue. This leads to better classifier weight estimates.

### 3.2 CONSISTENT INSTANCE CLASSIFICATION

We introduce a consistency loss to explicitly penalize the dissimilarity between augmented views from the same instance. Let $\text{sim}(\mathbf{u}, \mathbf{v}) = \mathbf{u}^\top \mathbf{v} / \|\mathbf{u}\|\|\mathbf{v}\|$ denote the inner product between $\ell_2$ normalized $\mathbf{u}$ and $\mathbf{v}$, i.e. cosine similarity. The consistency loss for two views $\mathbf{x}_n^i$ and $\mathbf{x}_n^j$ from the image $\mathbf{I}_n$ is formed as

$$\ell_c(\mathbf{x}_n^i, \mathbf{x}_n^j) = (1 - \text{sim}(\mathbf{x}_n^i, \mathbf{x}_n^j))^2 = (1 - \tilde{\mathbf{x}}_n^{i\,\top} \tilde{\mathbf{x}}_n^j)^2. \tag{2}$$

Here, we normalize the feature vector $\tilde{\mathbf{x}} = \mathbf{x}/\|\mathbf{x}\|_2$ as done in InstDisc and MoCo. The consistency loss for the $N$ images each with $K$ augmented views is written as

$$\mathcal{L}_c = \sum_{n=1}^{N} \sum_{i,j=1,i\neq j}^{K} \ell_c(\mathbf{x}_n^i, \mathbf{x}_n^j) = \sum_{n=1}^{N} \sum_{i,j=1,i\neq j}^{K} (1 - \tilde{\mathbf{x}}_n^{i\,\top} \tilde{\mathbf{x}}_n^j)^2. \tag{3}$$

The classification loss for the $N$ images each with $K$ augmented views is written as

$$\mathcal{L}_s = \sum_{n=1}^{N} \sum_{k=1}^{K} \ell_s(\mathbf{x}_n^k) = -\sum_{n=1}^{N} \sum_{k=1}^{K} \log \frac{e^{\mathbf{w}_n^\top \tilde{\mathbf{x}}_n^k / \tau}}{\sum_{j=1}^{N} e^{\mathbf{w}_j^\top \tilde{\mathbf{x}}_n^k / \tau}}. \tag{4}$$

where we let the classifier weight be an $\ell_2$-normalized vector: $\|\mathbf{w}\|_2 = 1$, which is similar to normalizing the prototype vector as done in InstDisc and MoCo.

We combine the two losses together,

$$\mathcal{L} = \mathcal{L}_s + \alpha \mathcal{L}_c, \tag{5}$$

where $\alpha$ is a weight for the consistency loss to balance the two losses, avoiding over-optimizing the consistency loss or merely optimizing the classification loss.

**Consistency helps optimizing the classification loss.** In general, when the features for each class are more compact, different classes are more easily separated and the softmax classification are more efficiently optimized. Our approach has the benefits: the feature distribution for the same instance is compact and the distributions for different instances are well separable, because of maximizing the consistency. Figure 2 (a) illustrates the benefit from simultaneously optimizing the consistency loss and the classification loss. Figure 2 (b) shows the insufficiency of only optimizing the classification loss. One can see that the distributions of different instances in Figure 2 (a) are better separated and the distribution for each instance is more compact.

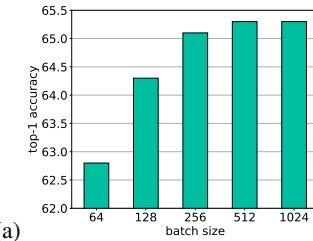 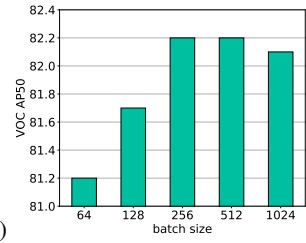 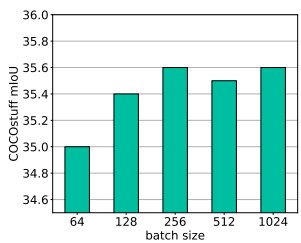

(a)                (b)                (c)

Figure 4: Performance with different batch sizes: (a) Linear evaluation on ImageNet, (b) VOC detection, and (c) COCO-stuff segmentation. The performances with batch sizes $256$, $512$ and $1024$ are similar.

Let's see how the consistency term makes the gradient of the classifier weight more effective. We have the gradient for the classification loss $\ell_s(\mathbf{x}_n^k)$ with respect to the classifier weight $\mathbf{w}_n$: $\frac{\partial \ell_s(\mathbf{x}_n^k)}{\partial \mathbf{w}_n} = (P_{nn}^k - 1)\tilde{\mathbf{x}}_n^k$, where $P_{nn}^k = \frac{e^{\mathbf{w}_n^\top \tilde{\mathbf{x}}_n^k / \tau}}{\sum_{r=1}^N e^{\mathbf{w}_r^\top \tilde{\mathbf{x}}_n^k / \tau}}$. The gradient from two views $\mathbf{x}_n^i$ and $\mathbf{x}_n^j$ is

$$\mathbf{g_w} = \frac{\partial \ell_s(\mathbf{x}_n^i)}{\partial \mathbf{w}_n} + \frac{\partial \ell_s(\mathbf{x}_n^j)}{\partial \mathbf{w}_n} = (P_{nn}^i - 1)\tilde{\mathbf{x}}_n^i + (P_{nn}^j - 1)\tilde{\mathbf{x}}_n^j. \tag{6}$$

According to the law of cosines, $\|\tilde{\mathbf{x}}_n^i\|_2 = 1$ and $\|\tilde{\mathbf{x}}_n^j\|_2 = 1$, we have

$$\|\mathbf{g_w}\|_2^2 = (P_{nn}^i - 1)^2 + (P_{nn}^j - 1)^2 + 2|(P_{nn}^i - 1)||(P_{nn}^j - 1)|\tilde{\mathbf{x}}_n^{i\top}\tilde{\mathbf{x}}_n^j. \tag{7}$$

When the consistency term is included, $\tilde{\mathbf{x}}_n^i$ and $\tilde{\mathbf{x}}_n^j$ are very close, implying that $\tilde{\mathbf{x}}_n^{i\top}\tilde{\mathbf{x}}_n^j$ is larger. In the case $P_{nn}^i$ and $P_{nn}^j$ are not changed, the magnitude $\|\mathbf{g_w}\|_2$ is larger, and accordingly the classifier weight $\mathbf{w}_n$ is updated effectively and quickly. In contrast, when the consistency term is not included, $\tilde{\mathbf{x}}_n^i$ and $\tilde{\mathbf{x}}_n^j$ might be very diverse as discussed (see the discussion in "**Optimizing the classification loss is not direct to optimize the consistency loss.**") This results in that $\|\mathbf{g_w}\|_2$ is smaller, and thus the classifier weight $\mathbf{w}_n$ is updated less effectively and less quickly.

Figure 3 shows the final classification loss $\mathcal{L}_s$ and consistency loss $\mathcal{L}_c$ value with different consistency loss weights. We can see that increasing the consistency weight when smaller than $2.5$ helps optimizing the classification loss, and when larger than $2.5$ harms the classification loss optimization. In Appendix B, we discuss the reason: over-weighting the consistency loss could lead to a trivial solution. In our experiments, we set $\alpha$ to $2.5$ in which case the training classification loss is minimum.

**Optimizing the classification loss is not direct to optimize the consistency loss.** Optimizing the classification loss intuitively expects that each instance lies in a different region. We expect that the augmented views of an instance $\mathbf{x}_n$ are assigned to the $n$th region and compactly distributed. We find that merely optimizing the classification loss $\mathcal{L}_s$ is not easy to make the features of the augmented views of the same instance contractive, consequently the features are not compactly distributed.

The reason is that larger similarity between augmented views is not explicitly encouraged, and is implicitly imposed through the classifier weight. The angle between $\tilde{\mathbf{x}}_n^i$ and $\tilde{\mathbf{x}}_n^j$, $\theta(\tilde{\mathbf{x}}_n^i, \tilde{\mathbf{x}}_n^j)$ (reflecting the similarity between $\tilde{\mathbf{x}}_n^i$ and $\tilde{\mathbf{x}}_n^j$, $\theta(\tilde{\mathbf{x}}_n^i, \tilde{\mathbf{x}}_n^j)$), is upbounded:

$$\theta(\tilde{\mathbf{x}}_n^i, \tilde{\mathbf{x}}_n^j) \leq \theta(\tilde{\mathbf{x}}_n^i, \mathbf{w}_n) + \theta(\mathbf{w}_n, \tilde{\mathbf{x}}_n^j). \tag{8}$$

Minimizing the classifier loss $\mathcal{L}_s$ if given $\mathbf{w}_n$, it is possible that the numerators (e.g, $\mathbf{w}_n^\top \tilde{\mathbf{x}}_n^i$ and $\mathbf{w}_n^\top \tilde{\mathbf{x}}_n^j$), are larger and accordingly the upbound $\theta(\tilde{\mathbf{x}}_n^i, \mathbf{w}_n) + \theta(\mathbf{w}_n, \tilde{\mathbf{x}}_n^j)$ is smaller. However, we find that there exist many transformations $\mathbf{R}$ so that the upbound is the same: $\mathbf{w}_n^\top (\mathbf{R}\, \tilde{\mathbf{x}}_n^i) = \mathbf{w}_n^\top \tilde{\mathbf{x}}_n^i$, and $\theta(\tilde{\mathbf{x}}_n^i, \mathbf{w}_n) = \theta(\mathbf{R}\tilde{\mathbf{x}}_n^i, \mathbf{w}_n)$. In this case, $\theta(\tilde{\mathbf{x}}_n^i, \tilde{\mathbf{x}}_n^j)$ is likely to be very different from $\theta(\mathbf{R}\tilde{\mathbf{x}}_n^i, \tilde{\mathbf{x}}_n^j)$. This implies that there is still a gap between optimizing the upbound $\theta(\tilde{\mathbf{x}}_n^i, \mathbf{w}_n) + \theta(\mathbf{w}_n, \tilde{\mathbf{x}}_n^j)$ and directly optimizing $\theta(\tilde{\mathbf{x}}_n^i, \tilde{\mathbf{x}}_n^j)$. As a result, merely optimizing the classification loss is not easy to make the features for one instance compactly distributed in the corresponding region.

**Batch size.** We present rough analysis showing that instance classification, including our approach, MoCo, and InstDisc, does not require large batch (see He et al. (2019) and the empirical validation in Figure 4 for our approach). We rewrite the loss function in Equation 5 as

$$\mathcal{L} = \sum_{n=1}^N [\alpha \sum_{i,j=1, i \neq j}^K (1 - \text{sim}(\mathbf{x}_n^i, \mathbf{x}_n^j))^2 + \sum_{k=1}^K \log \frac{e^{\mathbf{w}_n^\top \tilde{\mathbf{x}}_n^k / \tau}}{\sum_{j=1}^N e^{\mathbf{w}_j^\top \tilde{\mathbf{x}}_n^k / \tau}}]. \tag{9}$$

The reformulation indicates that the loss can be decomposed to the sum of components, where each component depends on a different instance. The separation between instances is got through the classifier weights each of which encodes the information of the corresponding instance. The decomposability property leads to that the optimization of $\mathcal{L}$ using SGD behaves similarly to the standard classification problem with SGD: large batch size is not necessary. Figure 4 shows that the performances with batch sizes 256, 512 and 1024 are similar.

In contrast, the contrastive loss over all the $N$ instances in SimCLR, is

$$\mathcal{L} = \sum\nolimits_{n=1}^{N} (\log \frac{e^{\tilde{\mathbf{x}}_n^{1\top} \tilde{\mathbf{x}}_n^2 / \tau}}{\sum_{j=1}^{N} e^{\tilde{\mathbf{x}}_n^{1\top} \tilde{\mathbf{x}}_j^1 / \tau} + e^{\tilde{\mathbf{x}}_n^{1\top} \tilde{\mathbf{x}}_j^2 / \tau}} + \log \frac{e^{\tilde{\mathbf{x}}_n^{2\top} \tilde{\mathbf{x}}_n^1 / \tau}}{\sum_{j=1}^{N} e^{\tilde{\mathbf{x}}_n^{2\top} \tilde{\mathbf{x}}_j^1 / \tau} + e^{\tilde{\mathbf{x}}_n^{2\top} \tilde{\mathbf{x}}_j^2 / \tau}}). \tag{10}$$

We can see that this can not be decomposed as a sum of components each of which depends on a different instance, which is a general requirement for SGD. Each instance depends on other instances. We believe that this is the reason why SimCLR needs large batch size (Chen et al., 2020b).

## 4 IMPLEMENTATION DETAILS

**Data augmentation.** We adopt the augmentation scheme similar to SimCLR (Chen et al., 2020b). We randomly crop the input image with the crop scale $(0.15, 1)$ and resize it to $224 \times 224$. Then we apply random horizontal flipping, color jittering, grayscale, and Gaussian blur.

**Network architecture.** We use ResNet-50 (He et al., 2016) to extract features. Following SimCLR we adopt the same projection head consisting of a two-layer batch-normalized MLP (Linear→BN→ReLU→Linear→BN) and reduce the feature dimension from 2048 to 128 in pretraining.

**Training.** We use the SGD algorithm with momentum optimizer. We set the momentum parameter to 0.9, the weight decay parameter to $1e-4$, the batch size to 512, and the epoch number to 200. We adopt the cosine learning rate schedule, with the initial learning rate 0.06. Each instance in the current mini-batch is augmented into two views during training. The temperature $\tau$ is set to be 0.1. We use SyncBN. For ablation study, we train all the models for 100 epochs. The training is performed on 8 NVIDIA V100 GPUs. We use the PyTorch 1.3 platform (Paszke et al., 2019) .

**Sampling classifier weight update.** The analysis is based on the standard SGD algorithm. For clarity, we assume each iteration samples 1 instance with two augmented views. The analysis can be easily extended to sampling more instances with more augmented views. The loss function becomes

$$\mathcal{L} = 2(1 - \text{sim}(\mathbf{x}_n^1, \mathbf{x}_n^2)) - \sum\nolimits_{k=1}^{2} \log \frac{e^{\mathbf{w}_n^\top \tilde{\mathbf{x}}_n^k / \tau}}{\sum_{j=1}^{N} e^{\mathbf{w}_j^\top \tilde{\mathbf{x}}_n^k / \tau}}. \tag{11}$$

It can be seen that the denominator in the second term on the right-hand side, $\sum_{j=1}^{N} e^{\mathbf{w}_j^\top \tilde{\mathbf{x}}_n^k / \tau}$, is a summation of $N$ elements, and thus the complexity is $\Theta(N)$. We propose to approximate it by summing fewer ($N' = 65536$ is the same to the queue size in MoCo He et al. (2019)) elements:

$$e^{\mathbf{w}_n^\top \tilde{\mathbf{x}}_n^k / \tau} + \beta \sum\nolimits_{j=1}^{N'} e^{\mathbf{w}_{s_j}^\top \tilde{\mathbf{x}}_n^k / \tau}, \tag{12}$$

where we let the sampling compensation weight $\beta = \frac{N-1}{N'}$ for a better approximation (See Appendix A.1). This approximation reduces the forward loss computation complexity to $\Theta(N')$.

The normal iteration process needs to update all the $N$ classifier weights in each iteration, implying the complexity is still $\Theta(N)$. Fortunately, through derivation (see Appendix A.2), we find that we do not need to compute the gradients and update the classifier weights corresponding to the instances that are not sampled. We can delay the update to the iteration that the instances are sampled. In other words, at each iteration we only compute the gradients and update the classifier weights corresponding to the instances that are sampled. Consequently, the gradient computation and the classifier weight update, and accordingly each iteration takes $\Theta(N')$ time.

In our implementation, rather than sampling all the $N'$ classifier weights at each iteration, we only use the classifier weights, which correspond to the (e.g., 512) instances in a mini-batch, to replace the classifier weights that are the earliest sampled. The potential benefit is to reduce the IO cost if we store the weights in the disk or the CPU memory and only store the sampled weights in the GPU memory, which is practically valuable for very large scale cases (e.g., $1B$ or more images).

Table 1: Illustrating how the consistency weight $\alpha$ influences the performance The observations are consistent to the one about the classification loss shown in Figure 3.

| | | VOC Det. | | COCO Det. | | COCO Keypoint | | DensePose | Instance Seg. | | Semantic Seg. | | | | |
|---|---|---|---|---|---|---|---|---|---|---|---|---|---|---|---|
| | | VOC07+12 | | Mask-R 1× | | Mask-R 2× | | DP-R | City. | LVIS | City. | Stuff | ADE | VOC | Context |
| $\alpha$ | LE | $AP^{bb}$ | $AP^{bb}_{50}$ | $AP^{bb}$ | $AP^{mk}$ | $AP^{kp}$ | $AP^{kp}_{50}$ | $AP^{dp}$ | $AP^{mk}$ | $AP^{mk}$ | mIoU | mIoU | mIoU | mIoU | mIoU |
| 0 | 63.6 | 54.9 | 80.7 | 38.4 | 34.9 | 65.8 | 87.2 | 63.7 | 31.6 | 24.2 | 76.7 | 33.4 | 41.5 | 75.2 | 47.2 |
| 0.5 | 65.0 | 56.1 | 81.5 | 39.0 | 35.3 | 66.2 | **87.3** | 64.5 | 32.4 | 24.7 | 76.9 | 33.6 | 41.0 | 76.3 | 33.6 |
| 1.5 | **65.1** | **56.7** | **82.0** | **39.3** | **35.5** | 66.3 | 87.1 | 64.4 | 32.3 | 24.4 | 77.3 | 34.3 | 41.7 | **77.2** | 48.7 |
| 2.5 | **65.1** | 56.4 | 81.9 | 39.2 | **35.5** | **66.3** | 87.1 | 64.4 | 33.1 | **25.4** | **77.9** | **35.3** | 41.7 | 77.1 | **49.4** |
| 3.5 | 64.6 | 56.6 | 81.6 | **39.3** | **35.5** | **66.3** | 87.2 | **65.0** | **33.2** | 24.9 | **77.9** | 34.5 | 42.0 | 76.9 | 48.8 |
| 4.5 | 64.7 | 56.1 | 81.8 | 39.2 | 35.4 | 66.0 | 87.2 | 64.4 | 32.6 | 24.6 | 77.0 | 34.3 | **42.1** | 77.0 | 48.9 |

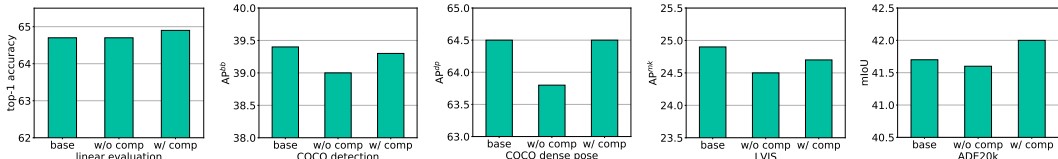

Figure 5: Illustrating the effect of sampling classifier weight update. Three results, the baseline w/o sampling, sampling update w/o sampling compensation, and sampling update w/ sampling compensation, are reported. The results show that sampling update w/ sampling compensation performs better than w/o sampling compensation and similar to the baseline w/o sampling.

## 5    EXPERIMENTS

We conduct the evaluation by training the models on ImageNet (Deng et al., 2009) w/o using the labels. We follow He et al. (2019) and adopt two protocols, linear evaluation on ImageNet, and downstream task performance with fine-tuning.

**Ablation study: consistency.** Figure 3 shows how the consistency weight parameter $\alpha$ influences the classification loss. The results are as our expectation and suggest that the classification loss decreases when the parameter $\alpha$ increases to a certain value 2.5 that choose to use in our experiments, and then the classification loss increases. The results on the downstream tasks and linear evaluation shown in Table 1 indicate consistent observations: the overall performance when $\alpha = 2.5$ performs satisfactorily, and better than the performance w/o consistency ($\alpha = 0$).

**Ablation study: sampling classifier weight update.** We evaluate how sampling classifier weight update and sampling compensation affect the performance. Figure 5 indicates that sampling compensation makes the results w/ the sampling scheme overall similar to the results w/o the sampling scheme and better than w/o sampling compensation ($\beta = 1$).

**Comparison with state-of-the-arts.** We compare our approach, consistent instance classification (ConIC) to recent state-of-the-art solutions: Exemplar-CNN, InstDisc, PIRL, MoCov1, MoCov2, AlignUniform, SwAV, and InfoMin[3]. The pretrained models of MoCo v1/v2, AlignUniform, SwAV, and InfoMin are obtained from GitHub provided by the corresponding authors. The PIRL pretrained model is obtained from PyContrast[4]. We implement Exemplar-CNN and InstDisc using the same setup with ours, including $\ell_2$ normalization and data augmentation. The comparison to these methods is fair as the models are pretrained with almost the same setting, e.g., #epochs is all 200, data augmentation is almost the same, the backbone is the same, and each instance is augmented to two views. We fine-tune all the models using the same setting for the downstream tasks.

The results for downstream tasks are given in Table 2. The overall performance of our approach (ConIC) is the best, and the overall performance of our approach w/ sampling classifier weight update (ConIC w/sampling) is the second best. In contrast, the best one among the previous methods, AlignUniform (Wang & Isola, 2020) performs satisfactorily for most tasks and unsatisfactorily for the segmentation tasks on Cityscapes, ADE20k, and Pascal-VOC. The superiority of our approach shows that minimizing the consistency loss improves the capability of characterizing the objects and the feature transferability.

---

[3]The results of other methods for downstream tasks are given in Appendix F.
[4]https://github.com/HobbitLong/PyContrast

Table 2: Comparison of our approach ConIC with recent state-of-the-art solutions. We highlight the best and second-best scores among the approaches w/o strong setup in red and blue, respectively.

| Method | Instance Seg. | | Semantic Segmentation | | | | | COCO Keypoint | |
|---|---|---|---|---|---|---|---|---|---|
| | City. | LVIS | City. | Stuff | ADE | VOC | Context | $AP^{kp}$ | $AP^{kp}_{50}$ |
| Exemplar-CNN (Dosovitskiy et al., 2016) | 32.4 | 25.3 | 77.1 | 34.0 | 41.5 | 77.0 | 48.6 | 66.1 | 87.2 |
| InstDisc (Wu et al., 2018) | 31.9 | 24.6 | 76.8 | 33.9 | 41.3 | 76.9 | 49.0 | 65.8 | 86.9 |
| PIRL (Misra & van der Maaten, 2019) | 32.2 | 25.0 | 75.4 | 34.1 | 40.2 | 75.3 | 47.6 | 66.2 | 86.9 |
| MoCo v1 (He et al., 2019) | 32.7 | 25.2 | 77.5 | 34.3 | 41.4 | 76.2 | 47.3 | 66.3 | 87.0 |
| MoCo v2 (Chen et al., 2020c) | 33.0 | 25.6 | 77.6 | 35.4 | 41.6 | 78.3 | 50.3 | 66.5 | 87.5 |
| AlignUniform (Wang & Isola, 2020) | 33.5 | 25.6 | 76.7 | 35.9 | 40.7 | 74.2 | 50.8 | 66.4 | 87.3 |
| ConIC | 33.6 | 25.6 | 78.5 | 36.0 | 42.1 | 78.9 | 50.9 | 66.5 | 87.3 |
| ConIC w/ sampling | 33.5 | 25.0 | 78.2 | 35.5 | 41.9 | 78.5 | 50.8 | 66.2 | 87.2 |
| *Approaches with stronger setup* | | | | | | | | | |
| SwAV (Caron et al., 2020) | 33.6 | 25.7 | 74.8 | 33.1 | 42.4 | 77.3 | 47.2 | 65.7 | 86.7 |
| InfoMin Aug. (Tian et al., 2020) | 33.7 | 25.6 | 78.2 | 36.2 | 42.3 | 78.7 | 51.1 | 66.5 | 87.5 |
| SimCLR Chen et al. (2020b) | 31.5 | 26.1 | 59.4 | 11.1 | 37.8 | 32.0 | 20.2 | 65.3 | 86.8 |

| Method | VOC Detection | | | | COCO Detection | | | | DensePose |
|---|---|---|---|---|---|---|---|---|---|
| | VOC07 | | VOC07+12 | | Mask-R 1× | | Mask-R 2× | | DP-RCNN |
| | $AP^{bb}$ | $AP^{bb}_{50}$ | $AP^{bb}$ | $AP^{bb}_{50}$ | $AP^{bb}$ | $AP^{mk}$ | $AP^{bb}$ | $AP^{mk}$ | $AP^{dp}$ |
| Exemplar-CNN (Dosovitskiy et al., 2016) | 47.1 | 75.0 | 53.9 | 80.1 | 38.7 | 35.0 | 41.4 | 37.4 | 64.4 |
| InstDisc (Wu et al., 2018) | 46.9 | 75.1 | 56.0 | 81.8 | 38.8 | 35.3 | 41.4 | 37.4 | 64.0 |
| PIRL (Misra & van der Maaten, 2019) | 45.9 | 73.9 | 55.4 | 81.0 | 38.7 | 35.1 | 41.4 | 37.4 | 64.3 |
| MoCo v1 (He et al., 2019) | 46.6 | 74.9 | 55.9 | 81.5 | 39.4 | 35.6 | 41.7 | 37.5 | 64.3 |
| MoCo v2 (Chen et al., 2020c) | 48.2 | 76.3 | 57.0 | 82.4 | 39.7 | 36.0 | 41.9 | 37.8 | 65.1 |
| AlignUniform (Wang & Isola, 2020) | 48.6 | 77.0 | 57.2 | 82.4 | 39.7 | 35.9 | 41.9 | 37.8 | 64.6 |
| ConIC | 48.8 | 76.8 | 57.5 | 82.4 | 39.9 | 36.0 | 41.9 | 37.9 | 64.9 |
| ConIC w/ sampling | 48.8 | 76.5 | 57.3 | 82.7 | 39.7 | 36.0 | 42.0 | 37.9 | 64.8 |
| *Approaches with stronger setup* | | | | | | | | | |
| SwAV (Caron et al., 2020) | 42.5 | 75.0 | 54.9 | 81.9 | 40.9 | 37.0 | 42.7 | 38.5 | 62.6 |
| InfoMin Aug. (Tian et al., 2020) | 48.6 | 77.0 | 57.6 | 82.7 | 40.6 | 36.7 | 42.5 | 38.4 | 65.6 |
| SimCLR Chen et al. (2020b) | 25.5 | 56.8 | 39.1 | 72.2 | 39.7 | 36.1 | 42.2 | 38.2 | 62.7 |

Table 3: Comparison for linear evaluation on ImageNet. Our approach gets comparable results to MoCov2, A-U, and others that train models using similar setup. See Appendix F for more discussions.

| Method | ConIC | ConIC-S | Local Agg. | E-CNN | InstDisc | PIRL | CMC | CPC v2 | MoCo v1 | MoCo v2 | SimCLR | A-U | PIC | PCL v1 | PCL v2 | BowNet | SeL-a | InfoMin | SwAV | SimCLR | BYOL |
|---|---|---|---|---|---|---|---|---|---|---|---|---|---|---|---|---|---|---|---|---|---|
| #Epochs | 200 | 200 | 200 | 200 | 200 | 200 | 200 | 200 | 200 | 200 | 200 | 200 | 200 | 200 | 200 | 280 | 280 | 800 | 800 | 1000 | 1000 |
| Top-1 acc | 67.6 | 67.4 | 60.2 | 64.4 | 65.1 | 61.7 | 66.2 | 63.8 | 60.6 | 67.5 | 66.6 | 67.7 | 67.6 | 61.5 | 67.6 | 62.1 | 68.8 | 73.0 | 75.3 | 69.3 | 74.3 |

In addition, we report the results of three approaches w/ stronger setup, InfoMin, SwAV, and SimCLR. We got the pretrained models provided by the authors. (1) InfoMin performs similarly to our approach, but it adopts stronger augmentation, RandomAugment (Cubuk et al., 2020) that is learned from supervised learning. (2) SimCLR (1000 epochs) performs inconsistently and surprisingly poorly[5]. (3) SwAV performs much better than our ConIC for COCO detection, and much worse for VOC detection, DensePose, semantic segmentation on Cityscapes, COCO stuff, Pascal-VOC, and Pascal-Context.

Linear evaluation results on ImageNet are in Table 3. Our approach performs competitively in comparison to MoCo v2, PIC, and PCL v2 whose training setup is similar to our approach. Other approaches, e.g, InfoMin, SwAV, BYOL, training the models using stronger augmentation, more views, more epochs, respectively, get higher performance. See more analysis in Appendix F.

## 6   CONCLUSION

We exploit the consistency loss minimization to help the optimization of the instance classification loss. The benefits include: the representations of different views of the same instance are more compact; the representations of different distances are more separable; the representations characterize more about the textured region in an image. These lead to high capability on downstream tasks like object detection and segmentation.

---

[5]We contacted the authors to see if we use the models correctly for some downstream tasks, and the feedback is they did not check the performance for those downstream tasks.

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

## A   SAMPLING CLASSIFIER WEIGHT UPDATE

### A.1   THE CHOICE OF $\beta$

We approximate the part (corresponding to the negative instances) of the denominator in the classification loss $\mathcal{L}_s = \sum_{k=1}^{2} \log \frac{e^{\mathbf{w}_n^\top \tilde{\mathbf{x}}_n^k / \tau}}{\sum_{j=1}^{N} e^{\mathbf{w}_j^\top \tilde{\mathbf{x}}_n^k / \tau}}$, by sampling a subset of classifier weights. That is

$$\beta \sum_{j=1}^{N'} e^{\mathbf{w}_{s_j}^\top \tilde{\mathbf{x}}_n^k / \tau} \approx \sum_{j=1, j \neq n}^{N} e^{\mathbf{w}_j^\top \tilde{\mathbf{x}}_n^k / \tau}. \tag{13}$$

Assume that the classifier weights corresponding to the $(N-1)$ negative instances, $\{\mathbf{w}_1, \ldots, \mathbf{w}_{n-1}, \mathbf{w}_{n+1}, \ldots, \mathbf{w}_N\}$, are i.i.d., we want that the expectations of the term on the left hand side and the right hand side are the same:

$$\mathrm{E}[\beta \sum_{j=1}^{N'} e^{\mathbf{w}_{s_j}^\top \tilde{\mathbf{x}}_n^k / \tau}] = \mathrm{E}[\sum_{j=1, j \neq n}^{N} e^{\mathbf{w}_j^\top \tilde{\mathbf{x}}_n^k / \tau}]. \tag{14}$$

The term on the left hand side is

$$\mathrm{E}[\beta \sum_{j=1}^{N'} e^{\mathbf{w}_{s_j}^\top \tilde{\mathbf{x}}_n^k / \tau}] = \beta N' \, \mathrm{E}[e^{\mathbf{w}^\top \tilde{\mathbf{x}}_n^k / \tau}], \tag{15}$$

where $\mathbf{w}$ has the same distribution with $\mathbf{w}_j$ $(j \neq n)$. Similarly, the term on the right hand side becomes

$$\mathrm{E}[\sum_{j=1, j \neq n}^{N} e^{\mathbf{w}_j^\top \tilde{\mathbf{x}}_n^k / \tau}] = (N-1) \, \mathrm{E}[e^{\mathbf{w}^\top \tilde{\mathbf{x}}_n^k / \tau}]. \tag{16}$$

Thus, we have $\beta = \frac{N-1}{N'}$. During the SGD iterations, the i.i.d. assumption does not hold. But our experiments show that the choice, $\beta = \frac{N-1}{N'}$, improves the performance, better than $\beta = 1$. We think that tuning $\beta$ manually might lead to superior performance.

### A.2   DELAY UPDATE OF THE UNSAMPLED CLASSIFIER WEIGHTS

Consider a classifier weight $\mathbf{w}$ that is sampled at the $(s)$th iteration and at the $(s+k)$th iteration, and is not sampled from the $(s+1)$th iteration to the $(s+k-1)$th iteration. We have (1) the gradient of the loss $\mathcal{L}$ with respect to $\mathbf{w}$ is zero, $\frac{\partial \mathcal{L}}{\partial \mathbf{w}} = \mathbf{0}$, (2) the gradient of the $\ell_2$ regularizer $\mathcal{R} = \frac{\lambda}{2}\|\mathbf{w}\|_2^2$ is $\frac{\partial \mathcal{R}}{\mathbf{w}} = \lambda \mathbf{w}$, and thus the gradient becomes: $\mathbf{g}_{\mathbf{w}^{(s)}} = \lambda \mathbf{w}$.

The update equation of SGD with momentum becomes

$$\begin{bmatrix} \mathbf{v}_{\mathbf{w}}^{(s+1)} \\ \mathbf{w}^{(s+1)} \end{bmatrix} = \begin{bmatrix} m & \lambda \\ -\eta^{(s+1)} m & (1 - \eta^{(s+1)} \lambda) \end{bmatrix} \begin{bmatrix} \mathbf{v}_{\mathbf{w}}^{(s)} \\ \mathbf{w}^{(s)} \end{bmatrix}, \tag{17}$$

from which we get:

$$\begin{bmatrix} \mathbf{v}_{\mathbf{w}}^{(s+k)} \\ \mathbf{w}^{(s+k)} \end{bmatrix} = \begin{bmatrix} m & \lambda \\ -\eta^{(s+k)} m & (1 - \eta^{(s+k)} \lambda) \end{bmatrix} \cdots \begin{bmatrix} m & \lambda \\ -\eta^{(s+1)} m & (1 - \eta^{(s+1)} \lambda) \end{bmatrix} \begin{bmatrix} \mathbf{v}_{\mathbf{w}}^{(s)} \\ \mathbf{w}^{(s)} \end{bmatrix}. \tag{18}$$

This means that we do not need to really compute $\mathbf{w}$ at the iterations in which it is not sampled, and only need to update it at the iteration in which it is sampled again,

In addition, we observe that $\mathbf{w}$ (unsampled) is updated independently and does not influence the update of other classifier weights. Consequently, we are safe to delay the update of the classifier weights that are not sampled to the iteration in which the weight is sampled again.

## B   MORE ANALYSIS

**Trivial solution for merely optimizing the consistency loss.** Let us look at the consistency loss in Equation 3. It is obvious that $\mathcal{L}_c \geq 0$. We can see that the minimum $\mathcal{L}_c = 0$ holds, if the features

of all the augmented views for an image are the same: $\tilde{\mathbf{x}}_n^i = \tilde{\mathbf{x}}_n^j$. It also holds in theory when that different images can have different representations: $\tilde{\mathbf{x}}_n^i \neq \tilde{\mathbf{x}}_m^j$. However, we empirically observe that merely optimizing the consistency loss always leads to the trivial solution: the representations of all the augmented views for all the images are the same, $\tilde{\mathbf{x}}_n^i \neq \tilde{\mathbf{x}}_m^j$.

**Hard sample mining.** It is known that the softmax loss has a benefit: hard samples contribute more to the gradient and thus the parameter update. We show that the consistence term has a similar property. The gradient of the consistence term $\ell_c$ in Equation 2 with respect to $\tilde{\mathbf{x}}_n^i$ is

$$\frac{\partial \ell_c}{\partial \tilde{\mathbf{x}}_n^i} = -2(1 - \tilde{\mathbf{x}}_n^{i \top} \tilde{\mathbf{x}}_n^j)\tilde{\mathbf{x}}_n^j. \tag{19}$$

In the hard sample case, the similarity $\tilde{\mathbf{x}}_n^{i \top} \tilde{\mathbf{x}}_n^j$ is smaller and far from 1, $(1 - \tilde{\mathbf{x}}_n^{i \top} \tilde{\mathbf{x}}_n^j)$ is larger, implying the gradient magnitude is larger. This means more contribution to the gradient. In the easy sample case, the contribution would be smaller.

## C  EVALUATION SETUP

### C.1  EVALUATION ON DOWNSTREAM TASKS

We perform object detection, COCO keypoint detection, COCO DensePose estimation and Instance segmentation experiments on Detectron2 (Wu et al., 2019) framework.

**Object detection.** We perform object detection on Pascal VOC (Everingham et al., 2010) and COCO (Lin et al., 2014) datasets. For Pascal VOC, we use Faster-RCNN (Ren et al., 2017) with R50-C4 backbone as the detector. Following He et al. (2019), extra BNs are added in newly initialized layers. We fine-tune all layers (including BN layers) in object detection experiments. Initial learning rate is 0.02. Two training schemes are adopted: (i) the model is trained on `train2007` set for 9k iterations, with learning rate decay at 6k and 8k iteration. (ii) the model is trained on `trainval07+12` set for 24k iterations, with learning rate decay at 18k and 22k iterations. We report $\text{AP}_{50}^{bb}$ and standard COCO-style $\text{AP}^{bb}$. For COCO object detection, we use Mask-RCNN (He et al., 2017) with R50-FPN (Lin et al., 2017) backbone as the detector. SyncBN is adopted in backbone, FPN and ROI Heads. The model is fine-tuned on `train2017` set and evaluated on `val2017` set. We use standard $1\times$ and $2\times$ fine-tune schedule. Standard COCO-style bounding box $\text{AP}^{bb}$ and mask $\text{AP}^{mk}$ are reported.

**COCO keypoint detection.** We perform human pose estimation on COCO keypoint (Lin et al., 2014) dataset. We use Mask-RCNN (He et al., 2017) (keypoint version) with R50-FPN backbone as the detector. SyncBN is adopted in backbone, FPN and ROI Head. The model is fine-tuned on `train2017` set and evaluated on `val2017` set. Standard $2\times$ fine-tune schedule is applied. We report $\text{AP}^{kp}$ and $\text{AP}_{50}^{kp}$.

**COCO DensePose estimation** For DensePose (Güler et al., 2018) estimation, We use Dense-Pose R-CNN with R50-FPN backbone. SyncBN is adopted in backbone, FPN and ROI Box Head. The model is trained on `train2014 + valminusminival2014` and evaluated on `minival2014`. We use "s1$\times$" fine-tune schedule (improved baseline "R_50_FPN_s1x" in Detectron2). We report $\text{AP}^{dp}$ of DensePose GPS metric.

**Instance segmentation.** We perform instance segmentation on COCO (Lin et al., 2014), Cityscapes (Cordts et al., 2016), and LVIS (Gupta et al., 2019) datasets. COCO instance segmentation is jointly-trained with COCO object detection with Mask-RCNN model. We use Mask-RCNN with R50-FPN for fine-tuning. SyncBN is adopted in backbone, FPN and ROI Heads. For Cityscapes, the model is trained on `cityscapes_fine_instance_seg_train` and evaluated on `cityscapes_fine_instance_seg_val` for 24k iterations. For LVIS, the model is trained on `lvis_v0.5_train` and evaluated on `lvis_v0.5_val` with $2\times$ schedule. Standard $\text{AP}^{mk}$ is reported.

**Semantic segmentation.** We perform semantic segmentation on Cityscapes (Cordts et al., 2016), COCO-stuff (Caesar et al., 2018), ADE20k (Zhou et al., 2017), Pascal-VOC (Everingham et al., 2010), and Pascal-Context (Mottaghi et al., 2014) datasets. We use DeeplabV3 (Chen et al., 2017)

with R50-dilated8 backbone. We use SGD with momentum optimizer and lambda poly learning rate schedule for semantic segmentation experiments. We employ cross entropy loss on both the final output of DeeplabV3 and the intermediate feature map output from stage3, where the weight over the final loss is 1 and the auxiliary loss is 0.4. Single-scale testing is adopted for all experiments. For Cityscapes experiments, we train the model for $40k$ iterations with batch size $8$, initial learning rate $0.01$, input size $1024 \times 512$. For COCO-stuff experiments, we train the model for $60k$ iterations with batch size 16, initial learning rate 0.01, input size $520 \times 520$. For ADE20k experiments, the model is trained for 150k iterations with batch size 16, initial learning rate 0.02, input size $520 \times 520$. For Pascal-VOC experiments, we use `train_aug2012 set` (augmented by Hariharan et al. (2011)) as training set. The model is trained for 60k iterations with batch size 16, initial learning rate 0.001 and input size $513 \times 513$. For Pascal-Context experiments, the model is trained for 30k iterations with batch size 16, initial learning rate 0.001 and input size $520 \times 520$. Standard mIoU metric is reported.

## C.2 LINEAR EVALUATION

We freeze the pretrained backbone and train a linear classifier on the frozen feature. The classifier is trained for 100 epochs with initial learning rate 75 and a cosine learning rate schedule. We set weight decay to 0. The data augmentation is the same as supervised ImageNet classification.

## D IMPLEMENTATION DETAILS OF THE TOY EXAMPLE

Figure 2 shows the learned feature distributions for 2D toy examples. We train the models (with a ResNet-50 encoder) on a toy dataset, containing 8 ImageNet images. We apply RandomCrop (0.7,1) on the images to generate augmented views. The models are trained for 200 epochs with a cosine schedule and initial learning rate 0.0001. We use batch size 8 and weight decay $1e - 6$. The dimension of features output from projection head is 2. For the classification only experiment, we set $\alpha = 0$. For the jointly optimization of the consistency loss and classification loss, we set $\alpha = 0.1$. After training, the learned features of 20 random augmented views of each image are recorded. We apply kernel density estimation with a Gaussian kernel of std $0.04$ on the recorded features for visualization. Each color represents the learned feature distribution of augmented views from an image.

## E DATA AUGMENTATION

We provide the PyTorch pseudo code of the data augmentation we adopted, as follows:

```
1   augmentation = [
2       transforms.RandomResizedCrop(224, scale=(0.15, 1.)),
3       transforms.RandomHorizontalFlip(),
4       transforms.RandomApply([
5           transforms.ColorJitter(0.8, 0.8, 0.8, 0.2)
6       ], p=0.8),
7       transforms.RandomGrayscale(p=0.2),
8       transforms.RandomApply([GaussianBlur([.1, 2.])], p=0.5),
9       transforms.ToTensor(),
10      normalize]
```

## F EXPERIMENT RESULTS OF OTHER METHODS

The abbreviations in Table 3 are explained in the following: ConIC-S = ConIC w/ sampling, Local Agg. = Local Aggregation (Zhuang et al., 2019). E-CNN = Exemplar-CNN (Dosovitskiy et al., 2016). A-U = AlignUniform (Wang & Isola, 2020). BowNet = Gidaris et al. (2020).

Table 4 shows the results of on VOC object detection form some other methods that are not included in Table 2. The results are got from the corresponding papers. BYOL, PCL, BowNet, SeLa adopted different evaluation setups and thus their results are not reported. Because of time limitation, cur-

rently we are not able to re-implement these algorithms or use their provided pretrained models and evaluate them on other downstream tasks.

Table 4: VOC object detection results for other methods that are not included in Table 2.

| Method | VOC Detection | | | |
| --- | --- | --- | --- | --- |
| | VOC07 | | VOC07+12 | |
| | $AP^{bb}$ | $AP^{bb}_{50}$ | $AP^{bb}$ | $AP^{bb}_{50}$ |
| Local Aggregation (Zhuang et al., 2019) | | 69.1 | | |
| PIC (Cao et al., 2020) | | | 57.1 | 82.4 |
| CPC v2 (Hénaff et al., 2019) (ResNet-161) | | 76.6 | | |
| ConIC | 48.8 | 76.8 | 57.5 | 82.4 |
| ConIC w/ sampling | 48.8 | 76.5 | 57.3 | 82.7 |

