# OpenReview forum: "Consistent Instance Classification for Unsupervised Representation Learning"
_ICLR.cc/2021/Conference — Reject_

### Official Review · AnonReviewer4 · 2020-10-27
**consistency loss improves instance classification; results on-par with existing methods**

**Rating:** 5
**Confidence:** 4

**Review:**

This paper proposes adding an additional loss term to instance classification, within the context of self-supervised pre-training.  Specifically, in addition to the standard classification loss term that views each image (and its augmentation) as a separate category, a second loss term is added, which is 1 minus the inner product of the representations for two augmentation views of the same image.  In a sense, this is incorporating an aspect of the contrasting learning approach, as two views of the same image are being explicitly pulled toward one another due to the consistency loss.

Positives:
+ sensitivity analysis with respect to the balancing coefficient $\alpha$ between the classification loss and consistency loss demonstrates that there is an optimal $\alpha$ such that the consistency loss leads to a minimum of the classification loss, even compared with setting $\alpha=0$.  In other words, the consistency loss is helping to optimize the classification loss.
+ related to above, empirical results and ablation study demonstrates the benefit of adding in consistency loss for subsequent downstream tasks
+ qualitatively, adding the consistency loss seems to help the network focus on relevant/textured regions of the images

Negatives:
- Novelty is arguably somewhat limited, as the proposed method boils down to adding a term that encourages similarity between two views of the same image, very similar to contrastive learning.
- Empirical results are mixed - in general on-par with existing methods.  Performance is generally consistent across the explored tasks, but does not appear to be significant better than existing approaches.  One data point that would be of interest is whether the proposed approach does better when training for more epochs, similar to methods such as SwAV or MoCo v2 on ImageNet linear evaluation.

Overall summary:

Given the negatives mentioned above, I currently see the paper as being borderline/marginally below threshold.

---

> ### Author Response · Authors · 2020-11-24
> **Response to AnonReviewer4**
>
>
> **Q1**: Novelty is arguably somewhat limited, as the proposed method boils down to adding a term that encourages similarity between two views of the same image, very similar to contrastive learning.
>
> *__A1__:
>  In terms of techniques, we agree that we do not to introduce new things: the instance classification formulation is studied before, the consistency loss is included in formulating the contrastive learning, and studied in other problems (we discussed the consistency loss in semi-supervised learning, for enforcing the similarity between the predictions/features of different views of the same unlabeled sample).*
>
> *Our novelty lies in: we explicitly point out that optimizing the instance classification loss with 1M categories for unsupervised representation learning is one of the key challenges, and we use the consistency term to ease the optimization.*
>
> **Q2**: Empirical results are mixed - in general on-par with existing methods. Performance is generally consistent across the explored tasks, but does not appear to be significant better than existing approaches.
>
> *__A2__:
>  Yes. The results are mixed. Our approach is more advantageous in the downstream tasks semantic segmentation with an absolute 0.9 IoU improvement on Cityscapes, and in other tasks, the improvements are relatively minor. However, we note that our approach performs consistently well across all downstream tasks in Table 2. On the contrary, most recent self-supervised learning approaches performs unstably unsatisfactorily on  various some downstream tasks.*
>
> **Q3**: One data point that would be of interest is whether the proposed approach does better when training for more epochs, similar to methods such as SwAV or MoCo v2 on ImageNet linear evaluation.
>
> *__A3__:
>  We do not have enough time to have the results for longer epochs, e.g., 1000 epochs. But we guess that our approach may gets improvements similar to the methods such as SwAV or MoCo v2 on ImageNet linear evaluation.*

---

### Official Review · AnonReviewer3 · 2020-10-28
**The contribution is not significant.**

**Rating:** 5
**Confidence:** 4

**Review:**

This paper studies the instance classification solution for an unsupervised representation learning problem. Particularly, this paper proposes an additional consistency loss that is simultaneously optimized with classification loss, in order to penalize feature dissimilarity between augmented views of the same instance. Such consistency loss makes classification loss optimization easier and avoids large batch size. Extensive experiments on downstream tasks, e.g., segmentation and detection, show the effectiveness of the proposed method.

Compared with previous instance classification solutions, the major contribution is that this paper introduces a consistency loss on augmented views from the same instance. The idea that compacts the augmented views is not new. And, the theoretical contribution of the proposed consistency loss is limited, which just penalizes the cosine dissimilarity. Thus, it is far from a "decent" ICLR paper.

---

> ### Author Response · Authors · 2020-11-24
> **Response to AnonReviewer3**
>
>
> **Q1**: Compared with previous instance classification solutions, the major contribution is that this paper introduces a consistency loss on augmented views from the same instance.
>
> *__A1__:
>  We want to clarify that our main contribution is more on showing that the consistency loss is helpful for optimizing the instance classification loss with 1M categories.*
>
> **Q2**: The idea that compacts the augmented views is not new.
>
> *__A2__:
>  We agree that compacting the augmented views is not new. Compacting the augmented views together with separating different instances are basic components, already used by previous instance classification and contrastive learning approaches, such as MoCo, SimCLR, and InstDisc.*
>
> *Our point DOES NOT lie in the idea of compacting the augmented views, and instead, that using an additional consistency loss (that is based on compacting the augmented views)to help optimize the instance classification loss, which is not studied before.*
>
> **Q3**: And, the theoretical contribution of the proposed consistency loss is limited, which just penalizes the cosine dissimilarity.
>
> *__A3__:
>  We do not want to argue if the theoretical contribution is limited. But, we want to point out that we provide the theoretical analysis in Section 3.2 to show how the consistency loss helps optimize the classification loss.*
>
> *One more point is that we explicitly point out that optimizing the instance classification loss with **1M categories** for unsupervised representation learning is one of the key challenges.*

---

### Official Review · AnonReviewer2 · 2020-10-28
**seems to be novel**

**Rating:** 5
**Confidence:** 3

**Review:**

The paper proposes a loss function for unsupervised representation learning. It has two terms: an instance classification loss and a consistency loss. The novel part seems to be the consistency loss. It explicitly penalizes the dissimilarity between different views of the same instance.

Good paper, accept

+The proposed method is supported with extensive experiments. The proposed consistency loss which is basically cos similarity between different views of the same sample seems to be novel.

-The proposed term L_C is kind of redundant. The proposed term L_C explicitly pushes positive samples to come closer based on cos similarity.
But, the other term, L_S, already bring positive samples closer and repels the negative samples. Here by positive, I mean samples that are different views of the same instance. Likewise, by negative I mean samples that are from different instances. That being said, explicitly adding L_c term with hyperparameter \alpha provided more fixability in balancing the push/repel phenomenon by tuning the hyper-parameter \alpha.

-Compared to the state-of-the-art methods reported in Table 2, the improvements are really really minor. In fact, in most columns, there is no meaningful improvement.

minor comment:
w/ stronger setup --> "with" stronger setup

---

> ### Author Response · Authors · 2020-11-24
> **Response to AnonReviewer2**
>
>
> Thanks for the position comments. Yes. The novel part lies in using the consistency loss for helping optimize the classification loss.
>
> **Q1**: The proposed term L_C is kind of redundant. The proposed term L_C explicitly pushes positive samples to come closer based on cos similarity. But, the other term, L_S, already bring positive samples closer and repels the negative samples. Here by positive, I mean samples that are different views of the same instance. Likewise, by negative I mean samples that are from different instances. That being said, explicitly adding L_c term with hyperparameter \alpha provided more fixability in balancing the push/repel phenomenon by tuning the hyper-parameter \alpha.
>
> *__A1__: This raised a good point.*
>
> *We agree that in some problems, L_C and L_S might be redundant. But, in our problem: instance classification with 1M categories, merely optimizing L_S is hard, and adding the consistency loss, L_C aims to help optimize the classification loss. It is not redundant in our application.*
>
> *We assume that by balancing the push/repel phenomenon (mentioned by you), it means to adjust the degrees of push and/repel provided in the classification loss L_S, i.e., the optimal parameters that minimizes  the classification loss is not good. In fact, we DO NOT aim to have a better balance, and instead, we still want to optimize the classification loss better and we pick the model parameters with smaller classification loss. In Figure 3, we show that when alpha < 2.5, the classification loss is optimized better than without using the consistency loss L_C. Experiment results also show that the model parameters with smaller L_S has better downstream performance.*
>
> **Q2**: Compared to the state-of-the-art methods reported in Table 2, the improvements are really really minor. In fact, in most columns, there is no meaningful improvement.
>
> *__A2__: Thanks for raising the issue. Our approach is more advantageous in the downstream task semantic segmentation with an absolute 0.9 IoU improvement on Cityscapes, and in other tasks, the improvements are relatively minor. We note that in Table 2, our improvements are consistent for most downstream tasks.*
>
> *More significantly, we demonstrate that one difficulty of the instance classification framework with 1M categories for unsupervised representation learning lies in the optimization side. We discussed this in Section 3.1 and the third paragraph in Section 1. MoCo uses the moving average scheme to compute the classifier weight estimate, easing the optimization.  And SimCLR, whose objective function might be better than instance classification, however, needs large batch size, which is discussed in the last part about “Batch size” in Section 3. With the consistency term, the instance classification loss is better optimized, and the learned representation achieves superior/comparable performance for downstream tasks compared to MoCo/SimCLR.*

---

### Decision · Program_Chairs · 2021-01-07
**Final Decision**

**Decision:**

Reject

**Comment:**

This paper introduces a consistency loss for instance discrimination by adding a term to maximize the squared dot product between two views of the same image. The impact of the proposed approach is evaluated on a variety of settings with mixed improvements. While reviewers generally found the proposed method to be interesting, there were concerns regarding the novelty of the approach, the size of the performance improvement, and the choice to focus on instance discrimination vs. more recent approaches based on contrastive instance discrimination.

While I do not share the reviewer concerns regarding novelty, I am sympathetic to the concerns regarding the size of the improvement and the focus on instance discrimination. As such, I recommend that the paper be rejected in its current form. I would encourage the authors to apply their analysis and method to more recent contrastive instance discrimination approaches such as SimCLR and SWaV as well as non-explicitly contrastive, but high performing methods like BYOL. I would also encourage the authors to provide quantitative empirical analyses demonstrating the impact of the consistency term on large models rather than just toy models to demonstrate the impact of the consistency term in representational space.